# Genome-Wide Association Analysis of Growth Traits in Hu Sheep

**DOI:** 10.3390/genes15121637

**Published:** 2024-12-20

**Authors:** Tingting Li, Feng Xing, Na Zhang, Jieran Chen, Yuting Zhang, Hengqian Yang, Shiyu Peng, Runlin Ma, Qiuyue Liu, Shangquan Gan, Haitao Wang

**Affiliations:** 1College of Life Science and Technology, Tarim University, Alar 843300, China; tingli0704@163.com (T.L.); xingfeng2001@126.com (F.X.); 2State Key Laboratory of Molecular Developmental Biology, Institute of Genetics and Developmental Biology, Chinese Academy of Sciences, Beijing 100101, China; zhangna4520@163.com (N.Z.); 17852401863@163.com (J.C.); zhangyut09@163.com (Y.Z.); yhq17806253397@163.com (H.Y.); pengsy@163.com (S.P.); rlma@genetics.ac.cn (R.M.); qyliu@genetics.ac.cn (Q.L.); 3College of Coastal Agricultural Sciences, Guangdong Ocean University, Zhanjiang 524088, China

**Keywords:** Ovine 40K SNP chip, GWAS, growth, candidate genes, Hu sheep

## Abstract

(1) Background: The Hu sheep is a renowned breed characterized by high reproduction, year-round estrus, and resistance to high humidity and temperature conditions. However, the breed exhibits lower growth rates and meat yields, which necessitate improvements through selective breeding. The integration of molecular markers in sheep breeding programs has the potential to enhance growth performance, reduce breeding cycles, and increase meat production. Currently, the applications of SNP chips for genotyping in conjunction with genome-wide association studies (GWAS) have become a prevalent approach for identifying candidate genes associated with economically significant traits in livestock. (2) Methods: To pinpoint candidate genes influencing growth traits in Hu sheep, we recorded the birth weight, weaning weight, and weights at 3, 4, 5, 6, and 7 months for a total of 567 Hu sheep, and genotyping was performed using the Ovine 40K SNP chip. (3) Results: Through GWAS analysis and KEGG pathway enrichment, we identified three candidate genes associated with birth weight (*CAMK2B*, *CACNA2D1*, and *CACNA1C*). Additionally, we found two candidate genes linked to weaning weight (*FGF9* and *BMPR1B*), with *CACNA2D1* also serving as a shared gene between birth weight and weaning weight traits. Furthermore, we identified eight candidate genes related to monthly weight (*FIGF*, *WT1*, *KCNIP4*, *JAK2*, *WWP1*, *PLCL1*, *GPRIN3*, and *CCSER1*). (4) Conclusion: Our findings revealed a total of 13 candidate genetic markers that can be utilized for molecular marker-assisted selection, aiming to improve meat production in sheep breeding programs.

## 1. Introduction

Sheep (*Ovis aries*) play a vital role in agriculture, providing not only milk, wool, and fur but also serving as an important source of meat for human consumption [1]. Within the context of meat sheep breeding, growth traits represent critical economic indicators influenced by environmental conditions, feeding practices, and, crucially, genetic factors, and genetic factors are significant determinants of sheep weight and meat yield [2]. Over centuries of domestication, sheep have experienced genetic variations that enable adaptation to diverse environments, resulting in modifications across various traits, including growth, reproduction, coat color, and milk production [3,4]. These genetic shifts have left distinct genomic imprints. For example, mutations in the *MSTN* gene have been linked to improved meat production [5,6], while the *FecB* mutation is associated with an increased number of lambs [7]. Similarly, mutations in the *ASIP* and *MC1R* genes affect wool color [8,9], and the *FGF5* mutation (known as the angora mutation) is associated with enhanced wool yield and quality [10]. Furthermore, mutations in the *KRTAP20* gene contribute to curly and wavy wool textures [11]. Collectively, these genetic variations have led to the diversification of sheep breeds. Understanding these genomic imprints is instrumental for revealing the genetic variations that influence sheep traits and can provide molecular markers for targeted genetic breeding aimed at improving sheep production.

Growth performance in sheep, encompassing parameters such as body weight, growth rate, and feed conversion efficiency, is predominantly affected by muscle cell proliferation, metabolic activity, and bone ossification rates. Genetic mutations that influence these biological pathways can drive variations in growth performance. Body weight, including birth weight, weaning weight, monthly weight, and slaughter weight, can serve as a critical indicator of growth performance. Growth traits are considered quantitative traits influenced by multiple genes, and quantitative trait loci (QTL) can elucidate some of this trait variation [12]. Despite the challenges and time constraints associated with direct selection breeding for quantitative traits, substantial genetic improvements can be realized through hybridization and genomic variations in key genes or mutations [13]. Genome-wide association studies (GWAS) represent a powerful methodology for identifying genomic loci significantly associated with quantitative traits. GWAS have been extensively utilized to pinpoint loci and functional genes correlated with various livestock traits, including reproduction [14,15,16], growth [17,18,19], horn type [20,21,22,23], and coat color [24,25,26], employing high-density single nucleotide polymorphism (SNP) panels and whole-genome resequencing technologies.

In recent years, notable advancements have been achieved in mapping candidate genes influencing sheep body weight through GWAS and other biological methodologies. A previous study involving 329 sheep from the Sunit, German Mutton, and Dorper breeds identified 36 candidate SNPs associated with body weight, chest girth, and shin circumference by utilizing the Illumina OvineSNP50 BeadChip for GWAS. The study highlighted several key genes, including *MEF2B*, *RFXANK*, *CAMKMT*, *TRHDE*, and *RIPK2*, some of which have also been validated in other breeds through selective sweep analysis [17,27]. GWAS focused on the body weight of 96 Baluchi sheep revealed that the *STRBP* and *TRAMIL1* genes were associated with birth weight, while *APIP* and *DAAM1* were linked to weaning weight; *PHF15*, *PRSS12*, and *MAN1A1* were linked to six-month weight; and *SYNE1*, *WAPAL*, and *DAAM1* were associated with yearling weight [28]. Furthermore, GWAS on carcass traits in 600 Scottish Blackface sheep identified several candidate genes affecting muscle, bone, and fat characteristics located on chromosomes 1, 3, 6, and 24, including *EFEMP1*, *FSHR*, *ABCG2*, *OST/SPP1*, *MAPK3*, and *TBX6* [18]. In studies involving the Illumina Ovine SNP 50K Bead Chip, candidate genes associated with production traits were identified in Qira Black sheep and German Merino sheep, with a total of 84 genes linked to traits such as body weight, height, body length, and chest girth [12]. Additionally, GWAS conducted on Akkaraman lambs using the Illumina Ovine SNP 50 Bead Chip focused on 90-day weaning weight and average daily gain (ADG) from 0 to 90 days. The results indicated that the rs427117280 polymorphism on chromosome 2 was significantly associated with these growth traits, with subsequent genotyping confirming that lambs with the GG genotype exhibited greater weight and weight gain compared to those with TG and TT genotypes [29]. For Hu sheep, genotyping with the Illumina Ovine SNP 50 Bead Chip revealed significant associations between two SNPs (76354330.1 T>A and s64890.1 G>A) on the X chromosome that were significantly associated with body weight. In addition, *CAPN6* showed significant differential expression in the biceps femoris and longissimus dorsi muscles of Hu sheep, making it a potential candidate gene affecting body weight [30]. These findings provided potential molecular markers for sheep genetic improvement, contributing to the enhancement of growth performance in Hu sheep through marker-assisted selection breeding.

In sheep breeding, major genes or SNP loci associated with growth traits can be identified at the genomic level and utilized as molecular markers. This study aims to identify significant genes and SNP loci that influence the body weight of Hu sheep at various growth stages. We continuously recorded the birth weight; weaning weight; and weights at 3, 4, 5, 6, and 7 months for a total of 567 Hu sheep. The Ovine 40K SNP chip along with GWAS were employed to identify candidate genes that may affect the body weight of Hu sheep across these different growth stages.

## 2. Materials and Methods

### 2.1. Experimental Animals

This study utilized Hu sheep sourced from Jiangsu Qianbao Animal Husbandry Co., Ltd. (Yancheng, China). A total of 567 healthy Hu sheep were monitored, with body weights recorded at key developmental stages: birth; weaning; and at 3, 4, 5, 6, and 7 months of age. Electronic scales were employed for accurate weight measurements. All experimental procedures adhered to the ethical standards set forth by the Animal Advisory Committee at the Institute of Genetics and Developmental Biology, Chinese Academy of Sciences (Approval No. AP2022015-C1).

### 2.2. Data Correction for Body Weight

To minimize the impact of varying measurement times and ages on the subsequent GWAS, the body weight data collected at different periods for the 567 Hu sheep were normalized to a consistent reference time point. Initially, a box plot was constructed to visualize the weight distributions, and outliers were identified and removed based on the interquartile range (IQR) (where IQR = Q3 − Q1). The upper limit was defined as Q3 + 1.5 *IQR, and the lower limit was defined as Q1 − 1.5 *IQR, where Q1 and Q3 represent the first and third quartiles, respectively. The weight data distribution of the Hu sheep at different ages before outlier removal is shown in Appendix A, respectively. We converted the measurement ages into weeks by dividing by 7 and calculated the average weights for different ages in months. The following formulas were used to estimate the weights of the rams and ewes:Ram: y = −0.028604x^2^ + 1.972782x + 1.817299
Ewe: y = −0.015625x^2^ + 1.558751x + 2.614646

In these formulas, y represents the mean weight, and x represents the age in weeks.

### 2.3. Genotyping and SNP Quality Control

In this study, blood samples were collected from the jugular veins of the Hu sheep and placed in anticoagulant tubes for DNA extraction. After extraction, DNA samples with A260/280 values between 1.8 and 2.0 were selected for quality control. Subsequently, all 567 samples were genotyped using an Ovine 40K SNP chip containing 41,456 SNPs (Neogen_China_CHN_OVNG50V02). VCFtools was employed to process the identified SNP sites, focusing exclusively on biallelic sites located on the primary chromosomes for subsequent analysis [31]. Furthermore, PLINK v1.9 [32] software was used for quality control with the following criteria: (1) removed SNPs with the minor allele frequency (MAF) < 0.05; (2) removed the SNPs of the site missing rate (MISS) > 0.5; (3) removed the individuals with missingness (MIND) > 0.5; (4) removed the SNPs of Hardy–Weinberg equilibrium (HWE) < 10^−6^. Ultimately, a dataset comprising 567 animals and 37,227 SNP sites was utilized for further analysis.

### 2.4. Principal Component Analysis

Principal component analysis (PCA) was conducted using GCTA software (version 1.90) to assess the genetic relatedness among individuals [33]. Additionally, PLINK software (version 1.9) was employed to visualize the PCA results, with the horizontal and vertical axes representing different principal components.

### 2.5. Genome-Wide Association Study

Following quality control (QC), a linear mixed model (LMM) was employed to perform association analysis on the 37,227 high-quality SNPs derived from 567 individuals using GEMMA software (version 0.98.1) [34]. The model can be articulated as follows: in the formula, y represents the target trait in this study, corresponding to n samples and n traits; W is the fixed effects matrix; α denotes the corresponding coefficients, including the intercept; x indicates the SNP genotype; β represents the effect size of the marker; u is a random effect; ε is the error; MVN_n_ denotes the multivariate normal distribution of the n-dimension; τ^−1^ is the variance of the residual errors; λ is the ratio of two variables; and K is the kinship matrix, with I_n_ as the identity matrix.
y = Wα + xβ + u + ε; u~MVN_n_ (0, λτ^−1^K), ε~MVN_n_ (0, λτ^−1^I_n_)

### 2.6. Gene Annotation

Gene annotation was performed using the Ovis aries genome (Oar_v3.1; GCF_000298735.1) as the reference, obtained from the National Center for Biotechnology Information (NCBI) databases (http://www.ncbi.nlm.nih.gov/, accessed on 17 August 2024). Candidate genes associated with body weight were identified within a 20 kb upstream or downstream region of significant SNPs [35], and the functional annotations of these genes were subsequently conducted.

### 2.7. Enrichment Analysis of Candidate Genes

Candidate genes significantly associated with body weight (*p* < 0.01) were selected based on the data from GWAS and gene annotation analyses. Enrichment analysis for these weight-related candidate genes was performed using the DAVID database (http://david.ncifcrf.gov/list.jsp, accessed on 8 September 2024), and the Kyoto Encyclopedia of Genes and Genomes (KEGG) pathways in which these genes were involved were identified. Additionally, the KEGG pathways were visualized using the Weishengxin website (https://bioinformatics.com.cn/, accessed on 5 October 2024).

## 3. Results

### 3.1. Descriptive Statistics of Quality Control and Phenotype

A total of 567 Hu sheep were genotyped using the Ovine 40K SNP chip, which includes 41,456 SNPs. After quality control, 37,227 high-quality SNPs were retained for analysis. The distribution of these SNPs across each chromosome is presented in Figure 1a and Appendix A. Statistical analysis showed that SNP detection rates for all chromosomes exceeded 90%, with the exception of the X chromosome (Chr 27).

Recordings were made for birth weight; weaning weight; and body weights at 3, 4, 5, 6, and 7 months for the 567 Hu sheep. The weight data exhibited an approximately normal distribution (Figure 1b–h). A PCA of the 37,227 SNPs from the 567 Hu sheep revealed no evidence of genetic stratification within the population (Figure 1i).

### 3.2. Genome-Wide Association Study

GWAS were performed using the 37,227 SNPs to identify associations with birth weight (at 0 day); weaning weight (at 50 days); and body weights at 3, 4, 5, 6, and 7 months of age in the 567 Hu sheep. The aim of this analysis was to pinpoint genomic regions or specific SNPs influencing these growth traits. Manhattan and Q_Q plots for body weight at various ages were shown in Figure 2a–g.

The GWAS results for birth weight and weaning weight did not reveal significant or consistent signals across the chromosomes. In contrast, the analysis for body weights at 3, 4, 5, 6, and 7 months demonstrated remarkable consistency, with significant correlations observed on chromosome 27 (Figure 2c–g). In addition, signals were also detected on chromosome 6, although they did not surpass the established significance threshold. Gene annotation and statistical analysis (*p* < 10^−3^) indicated that the significant region on chromosome 27 spans from 5.77 to 15.63 Mb and contains six notable loci. Among these, three loci are found in the intergenic region, while the remaining three loci are located within the genes *FIGF*, *KAL1*, and *LOC100535956*. The significant region on chromosome 6 extends from 34.18 to 50.84 Mb and encompasses the genes *CCESR1*, *KCNIP4*, *GPRIN3*, and *LOC105615454* alongside three additional loci situated in intergenic regions (Table 1). In addition to the loci identified on chromosomes 27 and 6, significant associations were discovered on other chromosomes, including 2, 9, and 15. These loci are linked to various genes, such as *WT1*, *WWP1*, *JAK2*, *PLCL1*, *LIN7B*, *NME1*, *IL1R1*, and *LOC105606489*, which may be associated with monthly weight in sheep (Table 1).

### 3.3. The KEGG Pathway Enrichment Analysis

To investigate and elucidate the biological pathways associated with body weight, enrichment analysis was conducted on candidate genes linked to birth weight, weaning weight, and monthly weight based on the GWAS results. A total of 355, 388 and 407 SNPs were identified (*p* < 0.01) for each trait, corresponding to 179, 193, and 232 genes, respectively. The analysis revealed significant enrichment in 19, 11, and 18 pathways for birth weight (Figure 3a), weaning weight (Figure 3b), and monthly weight (Figure 3c), respectively, with the most significantly enriched pathways and genes summarized in Figure 3 and Appendix A (*p* < 0.05).

The KEGG analysis indicated that the enrichment pathways associated with weight-related genes varied depending on the trait. Specifically, candidate genes linked to birth weight were primarily enriched in pathways such as axon guidance, oxytocin signaling pathway, circadian entrainment, and the wingless/integrated (Wnt) signaling pathway (Figure 3a). In contrast, genes associated with weaning weight were mainly enriched in pathways like taste transduction, Hippo signaling pathway, various types of N-glycan biosynthesis, and the calcium signaling pathway (Figure 3b). For monthly weight, the primary enrichment pathways included GnRH secretion, axon guidance, Hippo signaling pathway, and the MAPK signaling pathway, among others (Figure 3c).

## 4. Discussion

### 4.1. Genetic and Functional Insights into Birth and Weaning Weight in Sheep

The heritability of birth weight and weaning weight in sheep has been reported to range between 0.30 and 0.35 [36,37], highlighting the influence of both genetic and environmental factors on these traits. Genetic determinants include factors such as maternal heritability and sibling number, while non-genetic factors, including nutrition and management conditions, also significantly affect these traits [38,39]. One notable finding was the strong association between the polymorphism of SNP rs427117280 and pre-weaning growth [29], suggesting that this SNP could serve as a valuable genetic marker for early growth in sheep. Furthermore, two key SNPs located on chromosomes 10 and 13, within the *ATP8A2* and *PLXDC2*, have been identified as significant contributors to early growth traits [40]. GO and KEGG pathway analysis have further highlighted *ATP8A2* and *PLXDC2* as promising candidate genes influencing post-weaning weight. *ATP8A2* is involved in lipid translocation and is critical for maintaining lipid asymmetry in cellular membranes. While the precise function of *PLXDC2* remains unclear, it is implicated in cell proliferation and migration, and its expression may play a critical role in modulating growth-related traits in sheep.

In our study, KEGG enrichment analysis revealed that genes such as *CAMK2B*, *CACNA2D1*, and *CACNA1C* are significantly associated with birth weight in Hu sheep (Appendix A). These genes are involved in key signaling pathways, including axon guidance, oxytocin signaling, circadian entrainment, and Wnt signaling, all of which are crucial for growth and development. *CAMK2B* encodes a calcium/calmodulin-dependent protein kinase that plays a pivotal role in neurodevelopment. It contains a Ca^2+^ binding domain that regulates calcium influx, which is essential for synaptic plasticity [41]. Both *CACNA2D1* and *CACNA1C* are auxiliary subunits of voltage-gated calcium channels, critical for the density and function of high-voltage-activated calcium channels in the plasma membrane. These channels are key to neurotransmitter release and short-term synaptic plasticity. Mutations in these genes have been implicated in various neurological disorders [42,43]. Interestingly, in pigs (*Sus scrofa domestica*), *CACNA2D1* has been associated with reproductive and growth traits, including carcass composition [44]. Similarly, in cattle(*Bovine*), variations in *CACNA2D1* have been linked to carcass traits and meat quality [45,46]. These findings suggest that these calcium channel-related genes may also influence growth traits in sheep, potentially through the regulation of neurodevelopmental and cellular signaling pathways.

Our study also found that genes such as *CACNA1C* and *BMPR1B* along with the Hippo signaling pathway are significantly enriched in pathways associated with weaning weight (Figure 3b and Appendix A). The Hippo pathway is a conserved regulatory network controlling organ size and tissue homeostasis by regulating cell proliferation, apoptosis, and stem cell self-renewal [47,48]. Dysregulation of this pathway can lead to abnormal growth, organ hyperplasia, and tumorigenesis [49,50,51]. *BMPR1B*, a receptor in the bone morphogenetic protein (BMP) signaling pathway, is involved in skeletal development and bone homeostasis [52]. Disruption of *BMPR1B* signaling in mice(*Mus musculus*) leads to reduced bone mass and osteogenesis [53]. In sheep, the *FecB* mutation in *BMPR1B* has been associated with improved growth traits, including increased body weight and daily weight gain post-weaning compared to wild-type sheep [54]. Moreover, sheep with the *FecB* mutation in the *BMPR1B* gene had significantly lower body weights than those individuals with the AA genotype [55].

Furthermore, the involvement of genes such as *FGF9* and *CACNA1C*, which regulate growth and development, is consistent with their roles in cell proliferation and differentiation. In conclusion, these findings suggest that *BMPR1B*, *FGF9*, and *CACNA1C* are key candidate genes influencing weaning weight in sheep through their modulation of cellular and developmental pathways.

### 4.2. Genetic Variations and Pathway Analysis of Monthly Weight in Sheep

Our GWAS identified 14 SNPs significantly associated with monthly weight, located within eight genes: *FIGF*, *WT1*, *KCNIP4*, *JAK2*, *WWP1*, *PLCL1*, *GPRIN3*, and *CCSER1*. These genes are involved in several key biological processes, such as cell proliferation, skeletal development, and feed conversion efficiency.

The *FIGF* gene (also known as *VEGF-D*), located on chromosome 27, showed significant associations with body weights at 3, 4, 5, 6, and 7 months of age in the sheep. As a member of the VEGF family, *FIGF* plays a crucial role in angiogenesis and cell migration and has been shown to induce morphological changes in fibroblasts. In chickens, a polymorphism (C>G) in *FIGF* has been associated with variations in muscle fiber number, with chickens harboring the CC genotype exhibiting greater muscle fiber counts compared to those with the GC or GG genotypes [55]. *WT1* is a sequence-specific transcription factor involved in cell survival, apoptosis regulation, and differentiation. It has been shown to regulate mitotic spindle function during cell division. Disruption of *WT1* function leads to dysregulated cell growth, which is linked to developmental disorders and tumorigenesis [56,57]. Given its role in cell proliferation, *WT1* is likely to influence growth traits in sheep by regulating these fundamental processes. KCNIP4, a member of the voltage-gated potassium channel-interacting protein family, has been linked to body weight in multiple livestock species. In rabbits, GWAS identified *KCNIP4* as a key gene associated with body weight [58], while studies in chickens and cattle have also demonstrated its association with growth traits, including body weight at 8 weeks and weaning weight, respectively [59,60,61,62]. In sheep, the role of *KCNIP4* in regulating growth and muscle development functions was further supported [63]. These findings suggest that *KCNIP4* plays a vital role in the growth and development of various livestock species. JAK2, a non-receptor protein tyrosine kinase, plays an essential role in multiple cytokine and growth factor signaling pathways, including those involved in cell proliferation and centrosome amplification [64]. *JAK2* mediates the effects of growth hormone, erythropoietin, and several interleukins on cell survival and immune responses [65]. The GHR/JAK2 pathway is a critical signaling pathway for GH to promote cell proliferation and bone development, and studies have shown that this pathway induces DNA synthesis and proliferation in primary hepatocytes of adult rats [66]. However, inactivation of *JAK2* in knockout mice impaired growth and osteoclast function, leading to reduced bone resorption in the developing growth plate [67]. Additionally, *JAK2-KO* also caused CD4 T cell inactivation and inhibited the development and maturation of dendritic cells, thereby regulating T cell immune tolerance and affecting immune responses [68]. These findings highlight the critical role of *JAK2* in regulating growth and immune responses.

*WWP1*, a ubiquitin ligase belonging to the HECT family, acts as an inhibitor of bone growth. The expression level of *WWP1* increases in aging, which is associated with reduced bone mass without affecting bone morphology [69]. Its mechanism involves indirectly impacting the osteogenic differentiation capacity of bone marrow mesenchymal stem cells by degrading key osteogenic factors such as Runx2, thereby reducing the number of osteoblasts and the osteogenic process while having no effect on osteoclasts. Overexpression of *WWP1* results in reduced osteogenic differentiation, while knockout of *WWP1* enhances bone regeneration and osteoblast activity [69,70,71]. These findings suggest that *WWP1* may influence growth traits by modulating bone development and skeletal growth. *PLCL1*, also known as *PRIP*, is implicated in regulating hip size in humans and likely exerts its effects through modulation of the Ca^2+^ signaling pathway [72,73]. In *PLCL1* knockout mouse models, increased bone density and trabecular bone were observed along with reduced osteoclast differentiation capacity and diminished bone resorption. This suggests that *PLCL1* influences bone growth and differentiation by regulating intracellular Ca^2+^ concentration and the calcium–calcineurin–NFATc1 signaling pathway [74]. *GPRIN3* has been identified as a molecular marker for horse tendons [75], and functional studies in sheep have linked it to temperature adaptability, indicating that *GPRIN3* may have diverse biological roles and adaptive significance across species [76]. This underscores the complexity of gene functions in skeletal development.

The *CCSER1* has been shown to be associated with growth traits in different animal species. Specifically, *CCSER1* was found to influence growth performance in hybrid pig populations through GWAS analysis [77], and in goats, copy number variation in *CCSER1* was linked to growth traits in Chinese populations [78]. Moreover, *CCSER1* has been identified as a candidate gene for growth and feed efficiency in cattle [79,80]. In sheep, selection signals associated with adaptive and economically significant traits in 15 local Russian sheep breeds have pinpointed *CCSER1* as a key gene influencing growth traits [81,82]. *CCSER1* has been frequently identified and enriched in growth trait studies across pigs, cattle, goat, and sheep, suggesting that *CCSER1* may play a key role in livestock growth and feed efficiency.

## 5. Conclusions

In this study, genotyping was performed using the Ovine 40K SNP chip, followed by GWAS and KEGG enrichment analysis to investigate the genetic basis of body weight at different ages in 567 Hu sheep. The analysis identified three candidate genes associated with birth weight (*CAMK2B*, *CACNA2D1*, and *CACNA1C*), three genes linked to weaning weight (*FGF9*, *CACNA1C*, and *BMPR1B*), and eight potential candidate genes associated with monthly weight. These eight genes are implicated in a variety of biological processes, including cell proliferation and differentiation (*FIGF*, *WT1*, *KCNIP4*, and *JAK2*), bone growth and differentiation (*WWP1*, *PLCL1*, and *GPRIN3*), and feed conversion efficiency (*CCSER1*). These findings contribute to a deeper understanding of the genetic architecture underlying growth traits in Hu sheep, offering valuable insights into the molecular mechanisms that regulate body weight at different stages of development. The identified genes and pathways provide promising targets for future studies on improving growth performance and feed efficiency in sheep, with potential applications in breeding programs aimed at enhancing livestock productivity.

## Figures and Tables

**Figure 1 genes-15-01637-f001:**
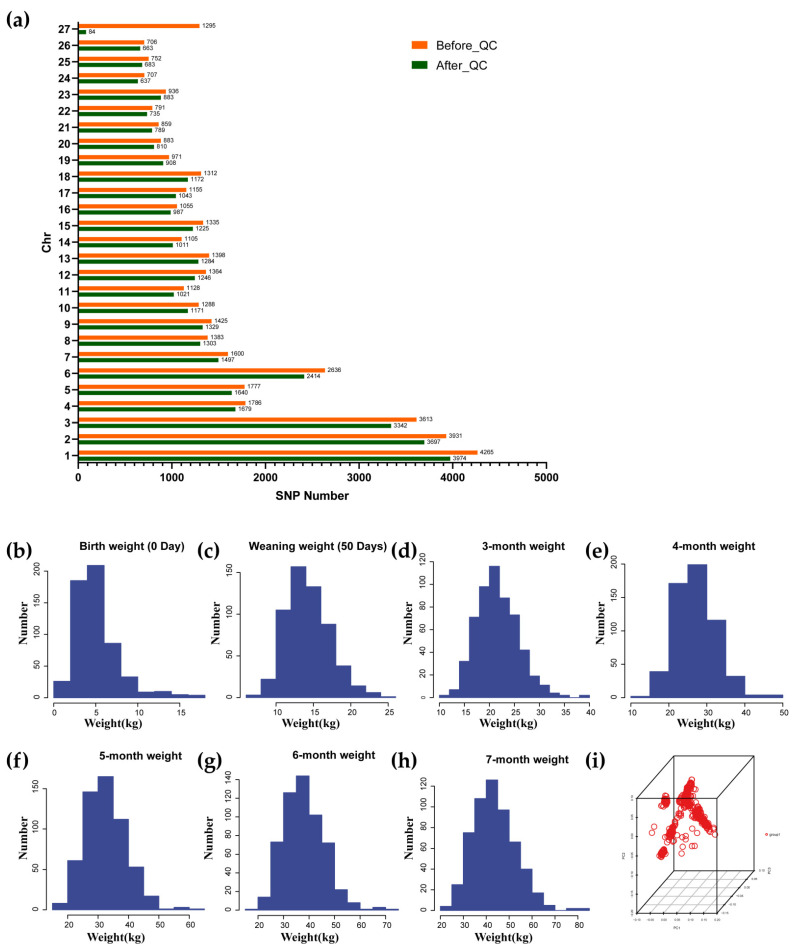
(**a**) SNP number of each chromosome before and after quality control. Chr: chromosome; Before_QC: SNP number of each chromosome before quality control; After_QC: SNP number of each chromosome after quality control; (**b**–**h**) Phenotypic distribution of the body weight measured at: (**b**) Birth weight; (**c**) Weaning weight; (**d**) 3-month weight; (**e**) 4-month weight; (**f**) 5-month weight; (**g**) 6-month weight; (**h**) 7-month weight; (**i**) PCA result of population stratification of Hu sheep.

**Figure 2 genes-15-01637-f002:**
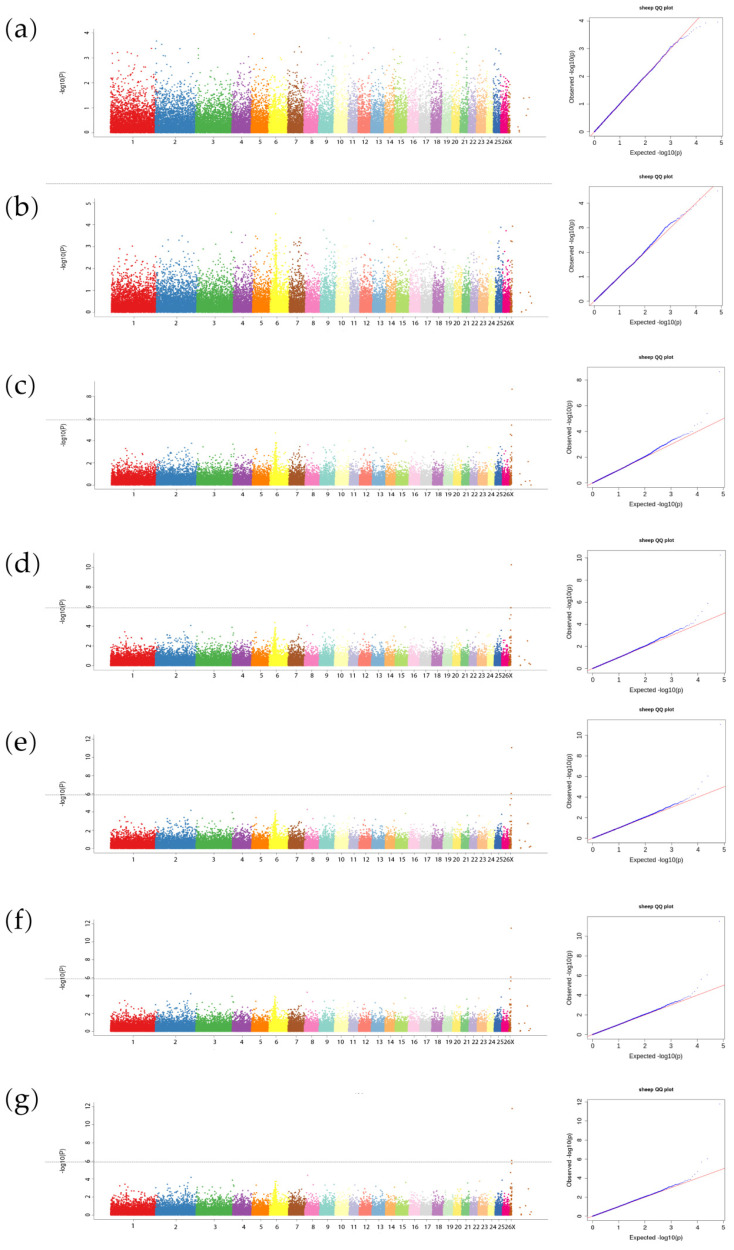
Manhattan plots of genome-wide association studies and corresponding Q_Q plots results for body weight of Hu sheep. The *x*-axis represents the chromosomes, and the *y*-axis represents the −log10 (*p*-value). Different colors indicate various chromosomes. The Q_Q plots show the observed vs. expected log *p*-value. (**a**) Birth weight; (**b**) Weaning weight; (**c**) 3-month weight; (**d**) 4-month weight; (**e**) 5-month weight; (**f**) 6-month weight; (**g**) 7-month weight.

**Figure 3 genes-15-01637-f003:**
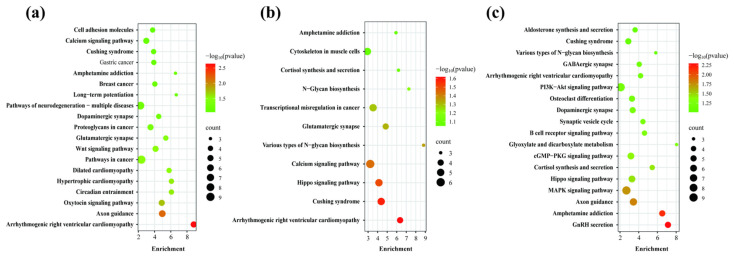
KEGG pathway for genes related to body weight. (**a**) Birth weight; (**b**) Weaning weight; (**c**) Monthly weight.

**Table 1 genes-15-01637-t001:** The top SNPs and candidate genes associated with monthly weight identified using a linear mixed model of Hu sheep.

Chr	Position	Ref/Mut	Gene	Location	*p*-Value (Monthly Weight)				
					3-Month	4-Month	5-Month	6-Month	7-Month
1	87,698,282	A/G	None	Intergenic region	5.10 × 10^−4^	3.65 × 10^−4^	3.45 × 10^−4^	3.56 × 10^−4^	3.76 × 10^−4^
2	72,815,358	G/A	*JAK2*	Intron3	1.95 × 10^−3^	7.90 × 10^−4^	5.21 × 10^−4^	4.24 × 10^−4^	3.81 × 10^−4^
2	175,670,044	G/A	None	Intergenic region	4.26 × 10^−4^	3.57 × 10^−4^	3.88 × 10^−4^	4.44 × 10^−4^	5.07 × 10^−4^
2	199,225,518	A/G	*PLCL1*	Intron1	6.48 × 10^−3^	1.53 × 10^−3^	6.50 × 10^−4^	3.79 × 10^−4^	2.64 × 10^−4^
2	217,936,417	G/A	None	Intergenic region	1.72 × 10^−4^	8.11 × 10^−5^	6.52 × 10^−5^	6.32 × 10^−5^	6.55 × 10^−5^
3	39,572,770	G/A	None	Intergenic region	3.57 × 10^−4^	3.80 × 10^−4^	4.74 × 10^−4^	5.94 × 10^−4^	7.23 × 10^−4^
3	223,215,094	A/G	None	Intergenic region	2.06 × 10^−4^	1.24 × 10^−4^	1.16 × 10^−4^	1.23 × 10^−4^	1.35 × 10^−4^
5	25,963,110	A/G	*LOC105615239*	Within	5.72 × 10^−4^	4.04 × 10^−4^	4.01 × 10^−4^	4.35 × 10^−4^	4.82 × 10^−4^
6	34,186,886	G/A	*CCSER1*	Intron1	4.82 × 10^−4^	5.14 × 10^−4^	6.07 × 10^−4^	7.17 × 10^−4^	8.31 × 10^−4^
6	34,448,315	G/A	*CCSER1*	Intron1	2.01 × 10^−5^	4.01 × 10^−5^	7.58 × 10^−5^	1.25 × 10^−4^	1.85 × 10^−4^
6	35,236,143	G/A	None	Intergenic region	6.75 × 10^−4^	6.30 × 10^−4^	6.89 × 10^−4^	7.75 × 10^−4^	8.68 × 10^−4^
6	35,503,461	C/G	*GPRIN3*	Intron1	1.46 × 10^−4^	1.54 × 10^−4^	1.99 × 10^−4^	2.58 × 10^−4^	3.25 × 10^−4^
6	35,511,899	A/G	*GPRIN3*	Exon2	2.79 × 10^−4^	3.35 × 10^−4^	4.58 × 10^−4^	6.10 × 10^−4^	7.77 × 10^−4^
6	35,512,157	C/G	*GPRIN3*	Exon2	2.79 × 10^−4^	3.35 × 10^−4^	4.58 × 10^−4^	6.10 × 10^−4^	7.77 × 10^−4^
6	35,512,755	A/G	*GPRIN3*	Exon2	2.79 × 10^−4^	3.35 × 10^−4^	4.58 × 10^−4^	6.10 × 10^−4^	7.77 × 10^−4^
6	35,554,875	A/G	*LOC105612138*	Within	1.42 × 10^−4^	2.09 × 10^−4^	3.15 × 10^−4^	4.45 × 10^−4^	5.86 × 10^−4^
6	35,562,563	A/C	*LOC105612138*	Within	2.64 × 10^−4^	4.20 × 10^−4^	6.44 × 10^−4^	9.06 × 10^−4^	1.19 × 10^−3^
6	36,864,972	A/G	*LOC105615454*	Within	1.86 × 10^−4^	2.98 × 10^−4^	4.65 × 10^−4^	6.61 × 10^−4^	8.67 × 10^−4^
6	36,886,733	C/G	*LOC105615454*	Within	1.86 × 10^−4^	2.18 × 10^−4^	2.82 × 10^−4^	3.58 × 10^−4^	4.37 × 10^−4^
6	40,570,462	A/T	*KCNIP4*	Intron1	3.93 × 10^−4^	6.09 × 10^−4^	9.13 × 10^−4^	1.26 × 10^−3^	1.61 × 10^−3^
6	41,047,416	A/G	*KCNIP4*	Intron1	1.56 × 10^−4^	1.35 × 10^−4^	1.46 × 10^−4^	1.65 × 10^−4^	1.86 × 10^−4^
8	16,967,044	A/G	*TBC1D32*	Intron22	2.32 × 10^−4^	8.34 × 10^−5^	5.28 × 10^−5^	4.29 × 10^−5^	3.91 × 10^−5^
9	33,108,286	A/G	None	Intergenic region	6.59 × 10^−4^	4.67 × 10^−4^	4.38 × 10^−4^	4.52 × 10^−4^	4.81 × 10^−4^
9	88,856,946	A/C	*WWP1*	Intron18	2.47 × 10^−4^	2.32 × 10^−4^	2.76 × 10^−4^	3.39 × 10^−4^	4.08 × 10^−4^
10	86,441,432	A/G	None	Intergenic region	1.00 × 10^−4^	2.37 × 10^−4^	4.63 × 10^−4^	7.60 × 10^−4^	1.10 × 10^−3^
11	35,133,720	G/A	*NME1*	Intron2	9.21 × 10^−4^	5.44 × 10^−4^	4.66 × 10^−4^	4.53 × 10^−4^	4.60 × 10^−4^
12	34,373,354	A/G	*LOC105606489*	Within	5.37 × 10^−4^	5.96 × 10^−4^	7.71 × 10^−4^	9.82 × 10^−4^	1.20 × 10^−3^
14	54,664,780	A/G	*LIN7B*	Intron1	1.68 × 10^−3^	9.96 × 10^−4^	8.45 × 10^−4^	8.12 × 10^−4^	8.15 × 10^−4^
15	61,369,364	G/A	*WT1*	Intron1	1.07 × 10^−4^	1.14 × 10^−4^	1.46 × 10^−4^	1.89 × 10^−4^	2.39 × 10^−4^
21	36,800,116	A/G	*LOC101122824*	Within	3.32 × 10^−4^	2.37 × 10^−4^	2.34 × 10^−4^	2.52 × 10^−4^	2.77 × 10^−4^
25	43,219,309	A/G	*ERCC6*	Exon5	5.91 × 10^−4^	2.55 × 10^−4^	1.75 × 10^−4^	1.46 × 10^−4^	1.34 × 10^−4^
27	5,938,597	A/G	None	Intergenic region	2.68 × 10^−5^	1.69 × 10^−5^	1.62 × 10^−5^	1.74 × 10^−5^	1.92 × 10^−5^
27	11,559,628	A/G	*LOC100535956*	Within	3.27 × 10^−4^	2.43 × 10^−4^	2.53 × 10^−4^	2.85 × 10^−4^	3.24 × 10^−4^
27	12,845,140	A/G	*FIGF*	Intron1	3.91 × 10^−6^	1.29 × 10^−6^	9.26 × 10^−7^	8.73 × 10^−7^	9.07 × 10^−7^
27	13,155,006	A/G	None	Intergenic region	3.35 × 10^−5^	6.82 × 10^−6^	3.37 × 10^−6^	2.39 × 10^−6^	1.99 × 10^−6^
27	15,633,576	A/G	None	Intergenic region	2.25 × 10^−9^	5.55 × 10^−11^	8.51 × 10^−12^	3.05 × 10^−12^	1.66 × 10^−12^

Chr: Chromosome; Ref/Mut: Reference/Mutation.

## Data Availability

The data presented in this study are available on request from the corresponding authors.

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
