# Peer review of "Genome-Wide Association Analysis of Growth Traits in Hu Sheep"

_genes, 2024, doi:10.3390/genes15121637_

Round 1

Reviewer 1 Report

Comments and Suggestions for Authors

Manuscript ID genes-3285701

In the manuscript entitled 'Genome-Wide Association Analysis of Growth Traits in Hu Sheep', the authors used the Ovine 40K SNP chip and GWAS to identify candidate genes that may influence the body weight of Hu sheep (n. 567) at different stages of their growth. The authors continuously recorded the birth weight, weight at weaning and weight at 3, 4, 5, 6 and 7 months of 567 Hu sheep. Using GWAS analysis and KEGG pathway enrichment, the authors identified 3 candidate genes (CAMK2B, CACNA2D1, CACNA1C) associated with birth weight, 3 candidate genes (FGF9, CACNA1C, BMPR1B) associated with weaning weight, and 8 candidate genes (FIGF, WT1, KCNIP4, JAK2, WWP1, PLCL1, GPRIN3, CCSER1) associated with monthly weight. The authors conclude that they have identified 14 candidate genetic markers for molecular marker-assisted selection to increase meat production in sheep breeding.

The manuscript is very interesting and in line with the aims of the Journal, it needs to be revised in the form of the "4. Discussion" section. Here are my observations, point by point:

Line 33: Sheep (Ovis aries) play a crucial role in meat supply for human all over the world.

Not only for the production of meat, but also for the production of milk and wool, expand this sentence.

Lines 37-39: During long-term domestication, sheep have undergone genetic variations to adapt to the environment, resulting in modifications in traits such as growth, reproduction, coat color, and resistance traits, which have imparted distinctive genomic imprints.

The sheep's coat (FLEECE) has two different types of hair: giarra (or goat hair), which comes from the main follicle, and wool, which comes from the secondary follicles. In addition, selection for wool has gradually led to the appearance of animals and breeds with mutations that favour the development of wool: the angora mutation, the mutation that suppresses the resting phase of the follicular cycle, the rex mutation, the merino mutation.

Reformulate the sentence

Lines 99-100: insert a clearly sentence “the aim of this study was/is ……

Lines 119-120: In these formulas, IQR refers to the interquartile range, Q1 is the first quartile, and Q3 is the third quartile. y represents the mean weight, and x represents the age in weeks.

In these formulas, IQR refers to the interquartile range, Q1 is the first quartile, and Q3 is the third quartile, y represents the mean weight, and x represents the age in weeks.

Line 163: 567 Hu sheep were genotyped using the Ovine 40K SNP chip which contains 41,456

Rewriting: "Five hundred and sixty-seven Hu sheep .....

Lines 237_240: Previous studies had reported that the heritability of birth weight and weaning weight in sheep ranges between 0.30 and 0.35[16, 17], indicating that these traits are influenced by genetic factors but are also affected by other factors, such as the number of siblings, the physical condition of the ewe, and the individual's nutritional status etc.

Rewrite the sentence more clearly

Line 255: and Wnt signaling.

Insert the acronym of Wnt

Lines 274-275: Malfunctions in this pathway can lead to excessive growth, tumour formation, and various diseases.

Contextualise the sentence with the citation.

Finally, my suggestion to the authors is to include the potential of the Hu breed of sheep for milk production, see the following references:

Xin Wang, Yancan Wang, Qiye Wang, Chunpeng Dai, Jianzhong Li, Pengfei Huang, Yali Li, Xueqin Ding, Jing Huang, Tarique Hussain & Huansheng Yang. (2023) The impact of early and mid-pregnant Hu ewes’ dietary protein and energy levels on growth performance and serum biochemical indicesJournal of Applied Animal Research 51:1, pages 174-181.

Zhang, X.; Liu, X.; Li, F.; Yue, X. The Differential Composition of Whey Proteomes in Hu Sheep Colostrum and Milk during Different Lactation Periods. Animals 202010, 1784. https://doi.org/10.3390/ani10101784

Comments on the Quality of English Language

A revision of the language is, in my opinion, necessary for the improvement of the paper.

Author Response

Comments 1: [Line 33: Sheep (Ovis aries) play a crucial role in meat supply for human all over the world. Not only for the production of meat, but also for the production of milk and wool, expand this sentence.]

Response1: This sentence has been changed to [Sheep (Ovis aries) is an important livestock, which can not only provide milk, fur and wool production for human beings, but also provide meat as food source]. Thank you for pointing this out. We agree with this comment. Therefore, we have changed this sentence, and this change can be found at page 1, lines 33-34, of the revised manuscript. 

Comments 2: [Lines 37-39: During long-term domestication, sheep have undergone genetic variations to adapt to the environment, resulting in modifications in traits such as growth, reproduction, coat color, and resistance traits, which have imparted distinctive genomic imprints.

The sheep's coat (FLEECE) has two different types of hair: giarra (or goat hair), which comes from the main follicle, and wool, which comes from the secondary follicles. In addition, selection for wool has gradually led to the appearance of animals and breeds with mutations that favour the development of wool: the angora mutation, the mutation that suppresses the resting phase of the follicular cycle, the rex mutation, the merino mutation.

Reformulate the sentence]

Response 2: This sentence has been changed to [During long-term domestication, sheep have undergone genetic variations to adapt to the environment, resulting in modifications in traits such as growth, reproduction, coat color, and milk production and quality[3, 4], which have imparted distinctive genomic imprints. For instance, the MSTN mutations have been shown to improve meat production in sheep[5, 6]; whereas the FecB mutation is associated with an increased number of lambs[7]; the mutations in the ASIP and MC1R genes lead to alterations in wool color[8, 9], the FGF5 mutation (angora mutation) is linked to enhanced wool yield and quality in sheep[10]. KRTAP20 gene mutation results in a curly, wavy wool[11]. These genetic variations have contributed to the diversification of sheep breeds.]. Thank you for pointing this out. We agree with this comment. Therefore, we have reformulated this sentence according to your suggestion, we also added some other sentence and references about major genes and their related different traits, and this change can be found at page 1-2, lines 38-47, of the revised manuscript

Comments 3: [Lines 99-100: insert a clearly sentence “the aim of this study was/is ……]

Response 3: This sentence has been changed to [The aim of this study is to identify significant genes and SNP loci that influence the body weight of Hu sheep at various growth stages...]. Thank you for pointing this out. We agree with this comment. Therefore, we have changed this sentence, and this change can be found at page 3, lines 99-100, of the revised manuscript

Comments 4: [Lines 119-120: In these formulas, IQR refers to the interquartile range, Q1 is the first quartile, and Q3 is the third quartile. y represents the mean weight, and x represents the age in weeks.

In these formulas, IQR refers to the interquartile range, Q1 is the first quartile, and Q3 is the third quartile, y represents the mean weight, and x represents the age in weeks.]

Response 4: [First, we generated a box plot of the weight data and removed outliers, defining the upper limit as Q3 + 1.5 * IQR and the lower limit as Q1 - 1.5 * IQR (where IQR = Q3 - Q1) (IQR refers to the interquartile range, Q1 is the first quartile, and Q3 is the third quartile.). Next, the measurement age was converted to weeks by dividing by 7, and the average weight for different ages in months was calculated. The formulas for calculating the weights of rams and ewes are as follows:

Ram: y= -0.028604x2+ 1.972782x+ 1.817299

Ewe: y= -0.015625x2+ 1.558751x+ 2.614646

In these formulas, y represents the mean weight, and x represents the age in weeks.] Thank you for pointing this out. We agree with this comment. Therefore, we have changed this position of “IQR refers to the interquartile range, Q1 is the first quartile, and Q3 is the third quartile”,Put it at the behind of “First, we generated a box plot of the weight data and removed outliers, defining the upper limit as Q3 + 1.5 * IQR and the lower limit as Q1 - 1.5 * IQR (where IQR = Q3 - Q1)”, and this change can be found at page 3, lines 118-119, of the revised manuscript

Comments 5: [Line 163: 567 Hu sheep were genotyped using the Ovine 40K SNP chip which contains 41,456

Rewriting: "Five hundred and sixty-seven Hu sheep....]

Response 5: This sentence has been changed to [Five hundred and sixty-seven Hu sheep were genotyped using the Ovine 40K SNP chip which contains 41,456]. Thank you for pointing this out. We agree with this comment. Therefore, we have changed this sentence, and this change can be found at page 4, line 169, of the revised manuscript

Comments 6: [Lines 237_240: Previous studies had reported that the heritability of birth weight and weaning weight in sheep ranges between 0.30 and 0.35[16, 17], indicating that these traits are influenced by genetic factors but are also affected by other factors, such as the number of siblings, the physical condition of the ewe, and the individual's nutritional status etc.

Rewrite the sentence more clearly]

Response 6: [Previous studies had reported that the heritability of birth weight and weaning weight in sheep ranges between 0.30 and 0.35[36, 37], indicating that these traits are influenced not only by genetic factors but also by other factors. Genetic factors include the number of siblings and maternal heritability, while other factors, such as feeding conditions and nutritional status, also play a significant role [38, 39].] Thank you for pointing this out. We agree with this comment. Therefore, we have changed this sentence, and this change can be found at page 9, lines 240-244, of the revised manuscript

Comments 7: Line 255: and Wnt signaling.

Insert the acronym of Wnt

Response 7: [Wingless / Integrated (Wnt) signaling]. Thank you for pointing this out. We agree with this comment. Therefore, we have Inserted the acronym of Wnt where it first appeared, and this change can be found at page 9, line 229, of the revised manuscript

Comments 8: Lines 274-275: Malfunctions in this pathway can lead to excessive growth, tumour formation, and various diseases.

Contextualise the sentence with the citation.

Response 8: [Malfunctions of this pathway can lead to organ excessive growth, tumor formation, and various diseases [49-51]]. Thank you for pointing this out. We agree with this comment. Therefore, we have added some citation for this sentence, and this change can be found at page 10, lines 279-280, of the revised manuscript

Comments 9: [Finally, my suggestion to the authors is to include the potential of the Hu breed of sheep for milk production, see the following references:

Xin Wang, Yancan Wang, Qiye Wang, Chunpeng Dai, Jianzhong Li, Pengfei Huang, Yali Li, Xueqin Ding, Jing Huang, Tarique Hussain & Huansheng Yang. (2023) The impact of early and mid-pregnant Hu ewes’ dietary protein and energy levels on growth performance and serum biochemical indices. Journal of Applied Animal Research 51:1, pages 174-181.

Zhang, X.; Liu, X.; Li, F.; Yue, X. The Differential Composition of Whey Proteomes in Hu Sheep Colostrum and Milk during Different Lactation Periods. Animals 202010, 1784. https://doi.org/10.3390/ani10101784]

Response 9: Thank for your suggestion. According to your suggestion, we have added the two references related to milk production in the Introduction section at line 40.

4. Response to Comments on the Quality of English Language

Point 1: A revision of the language is, in my opinion, necessary for the improvement of the paper.

Response 1: Thank you so much for your consideration of our manuscript. We recognized the importance of clear and effective communication in scientific writing. We have carefully reviewed the manuscript and made substantial revisions to improve the language. Specifically, we have addressed grammatical errors, improved sentence structure, and ensured that the terminology is precise and consistent throughout the document.

  1. Additional clarifications

None

Reviewer 2 Report

Comments and Suggestions for Authors

Dear Authors,

the aim of the manuscript could be of interest for the readers of this journal. However, it must be heavily improved. You need to improve the introduction and add several references for your sentences. I started to point out some limits in the introduction and discussion but I then gave up because the whole introduction must be rewritten and the discussion section must be checked and improved.

Lines 33-43: You have ten lines without a reference. 

Lines 44-55: please consider changing these lines; they seem disconnected from the other lines.

Lines 57-59: any references supporting this sentence? Quantitative traits can be (and have been) improved through "traditional breeding".

Lines 62-64: it is reductive to report just these few references since several manuscripts have been published about GWAS and genomic selection in sheep.

Lines 66-71: this not a scientific way to present a previous research. Please consider rephrasing as "A previous study on XXX sheep breed identified candidate genes..." or "XXX et al. [7] identified candidate genes associated with .... in the XXX sheep breed". 

Lines 112-114: why did you remove "outliers"?

Lines 115-118: why did you do this?

Lines 129-132: what about the call rate of animals?

Lines 134-137: the reference for GCTA is missing.

Line 145: what do you mean with "residual variable"?

Line 146: how did you compute the kinship matrix?

Line 150: "OAR3.1" is quite old. Was the same release as the SNP mapping? Why didn't you update the positions to the last available release?

Line 152: how did you decide to use 40kb? Any reference?

Lines 166-167: this is a very strange result; why did you have only 84 SNPs on CHR27? Which parameter (MAF, call rate, HWE) filtered out the markers? 

Lines 169-170: "The weight data conformed an approximately normal distribution" you cannot say that because you removed some records (what you named "outliers") without saying why you removed them.

Line 172: the quality of this figure needs to be improved.

Lines 181-182: above you mentioned also birth weight (0 days) and you have a plot for 0 days.

Line 200, Figure 2: why do you have some points outside the chromosomes? On the right of CHR 27.

Lines 231-234: these sentences are not well written and they can be omitted from the discussion.

Line 236: "GWAS results showed no continuous significant signals were detected across " this is not quite English.

Lines 237-241: references for the other factors affecting the traits? Why are you starting the discussion with heritability if you did not compute that?

Lines 355-356: "different animal breeds"... it's probably better to write "different animal species".

Lines 356: "CCSER1 is identified to related to growth performance "

Comments on the Quality of English Language

The English of the manuscript must be carefully checked and improved.

Author Response

1. Summary

2. Questions for General Evaluation

Reviewer’s Evaluation

Response and Revisions

Does the introduction provide sufficient background and include all relevant references?

Must be improved

This part had been improved in revision

Are all the cited references relevant to the research?

Must be improved

This part had been improved in revision

Is the research design appropriate?

Must be improved

This part had been improved in revision

Are the methods adequately described?

Must be improved

This part had been improved in revision

Are the results clearly presented?

Must be improved

This part had been improved in revision

Are the conclusions supported by the results?

Must be improved

This part had been improved in revision

Comments 1: [Lines 33-43: You have ten lines without a reference.]

Response 1: [Sheep (Ovis aries) is an important livestock, which can not only provide milk, fur and wool production for human beings, but also provide meat as source[1]. In the meat sheep breeding, growth trait is one of the economically significant indicators, they are not only influenced by the natural environment and feeding conditions but also significantly imp acted by genetic factors, which are key determinants of the weight and meat yield of sheep[2]. ......].Thank you for pointing this out. We agree with this comment. Therefore, we have added some references, and this change can be found at page1, lines 33-50, of the revised manuscript

Comments 2: [Lines 44-55: please consider changing these lines; they seem disconnected from the other lines.]

Response 2: Thank you for pointing this out. We agree with this comment that these sentences disconnected from the other lines (During embryonic development, mesodermal precursor cells differentiate into myogenic cells under the influence of surrounding tissue signals, this process requires the upregulation of factors such as Pax7, MyoD, and Myf5[1]. These proliferating precursor cells, known as myoblasts, exit the cell cycle and undergo a series of highly ordered programs to induce muscle-specific gene expression, ultimately forming mature muscle fibers [2]. On the other hand, the skeletal system, including bones, cartilage, tendons, and ligaments, originates from mesodermal mesenchymal stem cells and is regulated by various signaling pathways and transcription factors, differentiating into cells that constitute specific bone types. Pathways such as FGF, MAPK, and PI3K/Akt are involved in regulating chondrocyte proliferation and differentiation.). Therefore, we have deleted these lines, and this change can be found at page 2, between lines 50-51, of the revised manuscript

Comments 3: [Lines 57-59: any references supporting this sentence? Quantitative traits can be (and have been) improved through "traditional breeding".]

Response 3: [Growth traits are quantitative traits influenced by multiple genes, and quantitative trait loci (QTL) can explain part of the variation of traits [12]. Direct selection breeding for quantitative traits is both challenging and time-consuming, however, significant genetic improvements can be achieved through hybridization or genomic variations of important genes/mutations [13] ... ]. Thank you for pointing this out. Therefore, We have made changes based on your suggestions, not only by adding the reference but also by rewriting the sentence, and these changes can be found at page 2, lines 56-60, of the revised manuscript

Comments 4: [Lines 62-64: it is reductive to report just these few references since several manuscripts have been published about GWAS and genomic selection in sheep.]

Response 4: [GWAS has been widespreadly used to identify loci and functional genes significantly associated with livestock traits, including reproductive traits[14-16], growth traits[17-19], horn type[20-23], and coat color[24-26] using high-density SNP panels and whole genome resequencing technologies.]. Thank you for pointing this out. We agree with this comment. Therefore, We have added 3-4 references for every trait, and these changes can be found at page 2, lines 63-66, of the revised manuscript

Comments 5: [Lines 66-71: this not a scientific way to present a previous research. Please consider rephrasing as "A previous study on XXX sheep breed identified candidate genes..." or "XXX et al. [7] identified candidate genes associated with .... in the XXX sheep breed". ]

Response 5: [A previous study involving 329 sheep (Sunit, German Mutton, and Dorper sheep) identified 36 candidate SNPs associated with body weight, chest girth, and shin circumference using Illumina OvineSNP50 BeadChip to conduct GWAS. This study highlighted genes including MEF2B, RFXANK, CAMKMT, TRHDE, and RIPK2, some of which have also been validated in other breeds by using selective sweep analysis[17, 27]]. Thank you for pointing this out. We agree with this comment. Therefore, We have rephrased this sentence, and this change can be found at page 2, lines 69-73, of the revised manuscript

Comments 6: [Lines 112-114: why did you remove "outliers"?]

Response 6: Thank you for pointing this out. The outliers are abnormal data caused by a measurement error. The 3 standards deviation (3SD) method for identifying and removing outliers is a widely used and effective statistical approach. Under normal distribution conditions, approximately 99.7% of values are in the 3SD. Consequently, data out of this range are likely to be outliers and these data should be removed. 

Comments 7: Lines 115-118: why did you do this?

Response 7: By employing the 3SD method, those extreme values will not affect the trends of the data, made the results more reliable.

Comments 8: [Lines 129-132: what about the call rate of animals?]

Response 8: [(4) we removed animals with call rates missing at least 0.5]. Thank you for pointing this out. We agree with this comment. Therefore, We apologize for forgetting it, we have added it back now, and this change can be found at page 3, lines 135-136, of the revised manuscript

Comments 9: Lines 134-137: the reference for GCTA is missing.

Response 9: [GCTA software was used for Principal component analysis (PCA) in order to assess the genetic relatedness among individuals[33].] Therefore, We apologize for missing it, We have added it, and this supplement can be found at page 3, line 140, of the revised manuscript

Comments 10: [Line 145: what do you mean with "residual variable"?]

Response 10: -1 is the variance of the residual errors]. Thank you for pointing this out. I am sorry it is an error and it have been revised at revision, and this change can be found at page 4, lines 150-151, of the revised manuscript

Comments 11: [Line 146: how did you compute the kinship matrix?]

Response 11: Thank you for pointing this out. The kinship matrix was computed by GEMMA-gk, and the detailed process is shown in the following figure.

Comments 12: [ Line 150: "OAR3.1" is quite old. Was the same release as the SNP mapping? Why didn't you update the positions to the last available release?]

Response 12: The annotation points on the 40k chip we used were based on OAR_v3.1, this is a commercial chip from company (Neogen_China_CHN_OVNG50V02), the SNP mapping is same to this release. therefore, we did not update to the latest available release.

Comments 13: [Line 152: how did you decide to use 40kb? Any reference?]

Response 13: [Candidate genes for body weight were searched within 20kb the upstream or downstream of significant SNPs[35]]. Thank you for pointing this out. First of all, I am sorry this is an error, it should be 20kb instead of 40kb. In addition, The reference we decided to use, which is 20 kb, is: Gillen AE, Yang R, Cotton CU, et al. Molecular characterization of gene regulatory networks in primary human tracheal and bronchial epithelial cells. J Cyst Fibros. 2018;17(4):444-453. doi:10.1016/j.jcf.2018.01.009. This change can be found at page 4, line 158, of the revised manuscript

Comments 14: [Lines 166-167: this is a very strange result; why did you have only 84 SNPs on CHR27? Which parameter (MAF, call rate, HWE) filtered out the markers?]

Response 14: Thank for your question, through quality control, 1197 variants removed due to Hardy-Weinberg exact test. 14 variants removed due to minor allele threshold, so there were only 84 SNPs on CHR27.

Comments 15: [Lines 169-170: "The weight data conformed an approximately normal distribution" you cannot say that because you removed some records (what you named "outliers") without saying why you removed them.]

Response 15: The outliers refer to the data that were removed during the formulas-building process (Ram: y= -0.028604x2+ 1.972782x+ 1.817299, Ewe: y=-0.015625x2+1.558751x+ 2.614646), rather than belonging to the 567 samples we detected. The plot in Figure 1 is based on all 567 samples without removing any outliers, so we believe that the weight data conformed an approximately normal distribution.

Comments 16: Line 172: the quality of this figure needs to be improved.

Response 16: Thank you for pointing this out. We agree with this comment. We have increased the font size and enhancing the resolution of the exported file to improve clarity and quality.

Comments 17: [Lines 181-182: above you mentioned also birth weight (0 days) and you have a plot for 0 days.]

Response 17: Thank you for pointing this out. The born weight is 0-day weight, We have corrected the writing in the text and in Figure 1. This change can be found at page 6, line 188, of the revised manuscript

Comments 18: [Line 200, Figure 2: why do you have some points outside the chromosomes? On the right of CHR 27.]

Response 18: Thank you for pointing this out. There are some points look like in the outside of the chromosomes 27, but they are in CHR27, it's just farther away from clustered CHR27, as shown in the below figure.

Comments 19: [Lines 231-234: these sentences are not well written and they can be omitted from the discussion.]

Response 19: Thank you for pointing this out. We agree with this comment. Therefore, We have omitted it (In this study, we continuously measured the body weights of 567 Hu sheep from 0 to 7 months of age. Using an Ovine 40K SNP chip combined with GWAS analysis, we identified candidate genes or key SNP loci related to birth weight, weaning weight, and monthly weight in Hu sheep.), and this change can be found at page 9, between lines 238-239, of the revised manuscript

Comments 20: [Line 236: "GWAS results showed no continuous significant signals were detected across " this is not quite English.]

Response 20: Thank you for pointing this out. We agree with this comment. We believe this sentence is inaccurate. Therefore, we have decided to delete the paragraph without altering the overall meaning of the context.

Comments 21: [Lines 237-241: references for the other factors affecting the traits? Why are you starting the discussion with heritability if you did not compute that?]

Comments 22: [Lines 355-356: "different animal breeds"... it's probably better to write "different animal species".]

Response 22: [The CCSER1 gene has been shown to be associated with growth traits in different animal species...]. Thank you for pointing this out. We agree with this comment. We have changed it as your suggestion, this change can be found at page 11, line 363, of the revised manuscript

Comments 23: [Lines 356: "CCSER1 is identified to related to growth performance "]

Response 23: [CCSER1 is related to growth performance in a hybrid pig population through GWAS analysis[72]], the result of the sentence is from the paper of “Xue, Y.; Li, C.; Duan, D.; Wang, M.; Han, X.; Wang, K.; Qiao, R.; Li, X. J.Li, X. L. Genome-Wide Association Studies for Growth-Related Traits in a Crossbreed Pig Population. Anim Genet 2021,52,217-22”. In this study, the authors found that CCSER1 may be one of the important genes related to backfat thickness in pigs through GWAS analysis.

4. Response to Comments on the Quality of English Language

Point 1: The English of the manuscript must be carefully checked and improved.

Response 1: Thank you so much for your consideration of our manuscript. I agree that ensuring the English in our manuscript is of the highest quality is crucial. We have carefully reviewed the language and make necessary improvements to enhance clarity and readability.

5. Additional clarifications

None

Round 2

Reviewer 1 Report

Comments and Suggestions for Authors

Dear Authors,

The paper has been significantly improved according to the reviewers' suggestions.

Congratulations

Author Response

We greatly appreciate your consideration  and valuable suggestions of our manuscript once again.

Reviewer 2 Report

Comments and Suggestions for Authors

Dear Authors,

what you named "outliers" are not errors by default, they are errors only if they exceed the biological limit of the analyzed trait. Otherwise, they are just different values far from the average.

The English of the manuscript was not improved much.

Regarding the "20kb the upstream or downstream" (which has the in the wrong position) makes no much sense: 40kb is a very small region to account for LD.

Line 132: "we removed animals with call rates missing at least 0.5. " this sentence is not quite English. Moreover, it is the first time I see this very low threshold. Usually we want to consider animals that have at least 90-95% of the SNP.

I still do not understand why you used the equations at lines 118-121.  Even if weights are recorded at different time/age, you can simply add the age in the model to account for these differences. There is no genetic/statistical reasons to perform what you did in the paragraph 2.2. Moreover, you did not explain the reason why you did it, writing "... and standardised the data based on age" is not enough.

Comments on the Quality of English Language

The English of the manuscript is still a main limit.

Author Response

Comments 1: [what you named "outliers" are not errors by default, they are errors only if they exceed the biological limit of the analyzed trait. Otherwise, they are just different values far from the average.]

Comments 2: [The English of the manuscript was not improved much.]

Response 2: Thank you for pointing out issues with the English quality of our manuscript once again. We fully agree with your observation and apologize for the shortcomings in our previous revisions. We are committed to improving the English quality of the manuscript to meet the expected standards.

Comments 3: [Regarding the "20kb the upstream or downstream" (which has the in the wrong position) makes no much sense: 40kb is a very small region to account for LD.]

Response 3: Thank you so much for your consideration of our manuscript. We selected a distance of 20 kb upstream and downstream to annotate genes for the following reasons: (1) the linkage disequilibrium (LD) decay rate is significantly influenced by species. It is reported that LD in sheep persists for relatively shorter genomic distances than in cattle [1,2], pigs [3]or dogs [4]. In sheep, the LD decay rate is quite rapid, with a distance of 40 kb resulting in a reduction to less than half (Figure1) [5]. Therefore, 40 kb is a reasonable region to account for LD of sheep. 

(2) In addition, there is no standard range for gene annotation, and 20kb upstream or downstream of significant SNPs has been used in other articles [6]. We aim to identify the nearest genes associated with the significant SNPs, so we chose a distance of 40 kb to annotate genes.

Figure1 Average r2 values for each sheep population

References:

[1] Porto-Neto LR, Kijas JW, Reverter A. The extent of linkage disequilibrium in beef cattle breeds using high-density SNP genotypes. Genet Sel Evol. 2014;46:22. doi: 10.1186/1297-9686-46-22.

[2] Espigolan R, Baldi F, Boligon AA, Souza FR, Gordo DG, Tonussi RL, et al. Study of whole genome linkage disequilibrium in Nellore cattle. BMC Genomics. 2013;14:305. doi: 10.1186/1471-2164-14-305.

[3] Uimari P, Tapio M. Extent of linkage disequilibrium and effective population size in Finnish Landrace and Finnish Yorkshire pig breeds. J Anim Sci. 2011;89:609–614. doi: 10.2527/jas.2010-3249.

[4] Lindblad-Toh K, Wade CM, Mikkelsen TS, Karlsson EK, Jaffe DB, Kamal M, et al. Genome sequence, comparative analysis and haplotype structure of the domestic dog. Nature. 2005;438:803–819. doi: 10.1038/nature04338.

[5] Al-Mamun HA, Clark SA, Kwan P, Gondro C. Genome-wide linkage disequilibrium and genetic diversity in five populations of Australian domestic sheep. Genet Sel Evol. 2015;47:90. Published 2015 Nov 24. doi:10.1186/s12711-015-0169-6

[6] Gillen AE, Yang R, Cotton CU, et al. Molecular characterization of gene regulatory networks in primary human tracheal and bronchial epithelial cells. J Cyst Fibros. 2018;17(4):444-453. doi:10.1016/j.jcf.2018.01.009

Comments 4: [Line 132: "we removed animals with call rates missing at least 0.5. " this sentence is not quite English. Moreover, it is the first time I see this very low threshold. Usually we want to consider animals that have at least 90-95% of the SNP.]

Response 4: Thank you very much for considering our manuscript and for bringing this error to our attention. We sincerely apologize for any unclear responses or statements that may have caused confusion. Here are quality control steps that were conducted prior to GWAS: (1) removed SNPs with the minor allele frequency (MAF) < 0.05; (2) removed the SNPs of the site missing rate (MISS) > 0.5; (3) removed the individuals with missingness (MIND) > 0.5; (4) removed the SNPs of Hardy-Weinberg equilibrium (HWE) < 10-6. The missingness of individuals was highly correlated with the call rate of the animals, which was mentioned by your last review.

Due to the relatively low density of our chips and limited sample number, we established a more relaxed threshold in the first filter to retain as many individuals as possible. However, to ensure the quality of the data, we conducted a second round of filtering to confirm that call rate of the animals at least 90% of the SNPs. Therefore, we have corrected this sentence, and this change can be found at page 3, lines 139-142, of the revised manuscript.

Comments 5: [I still do not understand why you used the equations at lines 118-121. Even if weights are recorded at different time/age, you can simply add the age in the model to account for these differences. There is no genetic/statistical reasons to perform what you did in the paragraph 2.2. Moreover, you did not explain the reason why you did it, writing "... and standardised the data based on age" is not enough.]

Response 5: Thank you for pointing this out. Here we did not add age as a covariate in the analysis model of GWAS, but used the normalized weight data. Before conducting GWAS, five models have been tested for weight correction, and the current model we used in this manuscript has been proved to be the most reliable one. In this study, to mitigate the impact of varying measurement times/ages on weight, we first applied the correction model to normalize all weight data before conducting the GWAS analysis, which is also reported in other study[1]. We appreciate your suggestion and the method you suggested will be taken into consideration in our future analyses. 

Moreover, we have explained why we did this in revised manuscript, it is [To minimize the impact of varying measurement times and ages on the subsequent GWAS, the body weight data collected at different periods for 567 Hu sheep were normalized to a consistent reference time point.], and this change can be found at page 3, lines 119-121, of the revised manuscript.

[1] Khazaei-Koohpar H, Gholizadeh M, Hafezian SH, Esmaeili-Fard SM. Weighted single-step genome-wide association study for direct and maternal genetic effects associated with birth and weaning weights in sheep. Sci Rep. 2024 Jun 7;14(1):13120.

4. Response to Comments on the Quality of English Language

Point 1: [Comments on the Quality of English Language: The English of the manuscript is still a main limit.]

Response 1: Thank you very much for your thorough review of our manuscript and for highlighting the issue with the quality of English language. We agree your viewpoint and acknowledge that the clarity and professionalism of the language used is crucial for the impact and acceptance of the work. Therefore, professional English editing has been performed to improve my manuscript to ensure the precision and fluency of the language. Thank you again for your valuable suggestions and time.

5. Additional clarifications

None

Round 3

Reviewer 2 Report

Comments and Suggestions for Authors

na

Author Response

Dear reviewer,

Thank you for your kind attention and review of our manuscripts. We sincerely apologize for not describing some issues clearly in previous reply and revision. We would like to make these points more clearly. To these issues, the responses are as follows,

1. It is suggested to utilize the complete phenotypic data to create visualizations, which would allow for a more straightforward identification of outliers.

Thank you for your suggestion. In our study, we used the 3 standards deviation (3SD) method for identifying and removing outliers ,and it is a widely used and effective statistical approach. Under normal distribution conditions, approximately 99.7% of values are in the 3SD [1]. Consequently, data out of this range are likely to be outliers and these data should be removed.

In our study, we do utilize the complete phenotypic data to create visualizations and outliers elimination. However, only 567 samples from these individuals were genotyped using an Ovine 40K chip, and the corrected body weights of these individuals were utilized as phenotypic data for GWAS. The distribution of the 567 samples data is visualized in Figure 1(b-h) of the manuscript.

[1] Azadinia B, Khosravinia H, Masouri B, Kavan BP. Effects of early growth rate and fat soluble vitamins on glucose tolerance, feed transit time, certain liver and pancreas-related parameters, and their share in intra-flock variation in performance indices in broiler chicken. Poult Sci. 2022;101(5):101783.

2. The images mentioned by the author in their responses were not located. Specifically:

(1) The kinship matrix was computed by GEMMA-gk, and the detailed process is shown in the following figure.

Comments 11: [Line 146: how did you compute the kinship matrix?]

Response 11: Thank you for pointing this out. The kinship matrix was calculated by GEMMA-gk, This data is PLINK binary format in our study, the basic usage of which is ‘./gemma -bfile [prefix] -gk [num] -o [prefix]’. The detailed process is shown in the following figure 1. This method is also applied in other articles[1-2].

Figure 1 The computer process of kinship matrix

[1] Lee JB, Kang YJ, Kim SG, et al. GWAS and Post-GWAS High-Resolution Mapping Analyses Identify Strong Novel Candidate Genes Influencing the Fatty Acid Composition of the Longissimus dorsi Muscle in Pigs. Genes (Basel). 2021;12(9):1323.

[2] Zhou X, Stephens M. Efficient multivariate linear mixed model algorithms for genome-wide association studies. Nat Methods. 2014;11(4):407-409.

(2)There are some points that appear to be outside chromosome 27, but they are actually within CHR27; they are just farther away from the clustered region of CHR27, as shown in the below figure.

Comments 18: [Line 200, Figure 2: why do you have some points outside the chromosomes? On the right of CHR 27.]

Response 18: Thank you for pointing this out. It has been brought to our attention that some data points appear to be positioned outside of Chromosome 27. However, these points are indeed in Chromosome 27. They are located at a greater distance from the main cluster of Chromosome 27, which can be observed in the below figure. The data in the red box in the below figure 2 provide the detailed location information of these points.This spatial arrangement does not imply any discrepancy in the data.

Figure 2 The detailed location information for points located away from the main cluster on Chromosome 27

3.The reviewer has repeatedly pointed out issues with the manuscript's language. The author is requested to revise it thoroughly.

Thank you for pointing out issues with the English quality of our manuscript once again. We fully agree with your observation and apologize for the shortcomings in our previous revisions. Therefore, professional English editing has been performed to improve our manuscript to ensure the precision and fluency of the language. We apologize that the submitted revised manuscript did not include any edited markers due to the extensive modifications made to the language content. However, we promised that a thorough revision has been conducted. Please review it again. Thank you again for your valuable suggestions and time.

 4.Two issues appear to have been left unaddressed by the author:

(1) Comments 21: [Lines 237-241]: references for the other factors affecting the traits? Why are you starting the discussion with heritability if you did not compute that?

Response21: Firstly, Other factors affecting the traits mainly refer to non-genetic factors,including nutrition and management conditions, and references as follow: 

[1] Ghasemi, M.; Zamani, P.; Vatankhah, M.Abdoli, R. Genome-Wide Association Study of Birth Weight in Sheep. Animal 2019,13,1797-803.

[2] Zamani, P.Mohammadi, H. Comparison of Different Models for Estimation of Genetic Parameters of Early Growth Traits in the Mehraban Sheep. J Anim Breed Genet 2008,125,29-34.

Additionally, We discussed heritability to illustrate that birth weight and weaning weight are traits with moderate heritability. Consequently, these traits can be influenced by both genetic and non-genetic factors. While heritability has been calculated for our population of Hu sheep (unpublished data), this study focuses on identifying important genes and SNPs that affect growth, so we do not elaborate it.

(2) What you named "outliers" are not errors by default; they are errors only if they exceed the biological limit of the analyzed trait. Otherwise, they are simply values far from the average.

Response: Thank you for pointing this out. We apologize for our inaccurate description in our previous revision. We would like to clarify that the term "outliers" does not automatically imply errors in our dataset. The term "outliers" in our manuscript refers to data which lie outside the range of Q1 - 1.5 * IQR and Q3 + 1.5 * IQR. In order to ensure the quality of the analytical data, we consider that data conforming to a normal distribution could be used for downstream analyses. Any values that fall outside of the range will be classified as outliers and removed.